



# Evidence of coastal trapped wave scattering using high-frequency radar data in the Mid-Atlantic Bight

Kelsey Brunner[1] and Kamazima M. M. Lwiza[1]

[1]Stony Brook University, School of Marine and Atmospheric Sciences, Stony Brook University, Stony Brook, New York, USA

**Correspondence:** Kelsey Brunner (keliz.brunner@gmail.com)

**Abstract.** Coastal trapped waves (CTWs) become scattered when they encounter irregular coastlines and bathymetry during propagation. Analytical and modeling studies have provided some information about the different types of shelf geometries that can induce scattering, but much of the CTW scattering process generally remains a large knowledge gap. Furthermore, CTW scattering has never before been directly identified with observations. High-frequency radar surface velocity data covering

the Mid-Atlantic Bight (MAB) continental shelf provides unprecedented observations of CTWs within a region with a highly complex coastline and bathymetry. A combination of velocity vector maps from real vector empirical orthogonal function (R-EOF) analysis and phase maps from complex empirical orthogonal function (C-EOF) analysis allow the identification of CTW scattering by assuming each EOF mode corresponds to a CTW mode. Abrupt jumps in phase in association with magnitude amplification/reduction or directional rotation of velocity vectors are indications of scattering. Using these guidelines, Georges

Bank, Hudson Shelf Valley, Delaware Bay mouth, Chesapeake Bay mouth, and the North Carolina shelf are identified as high scattering regions within the MAB. Furthermore, stratification is confirmed to increase scattering into progressively higher order modes through a cascading process by comparing winter and summer cases, which supports previous theoretical and numerical model predictions. The simple methodology used here can be applied to observations of CTWs on other coastlines around the world to identify additional scattering regions and help close the knowledge gap.

## 1 Introduction

Coastal trapped waves (CTWs) were first observed around the coast of Australia in the 1960's (Hamon, 1962, 1966). Many observational and theoretical studies on the topic were conducted throughout the 1970's and 1980's (notably Buchwald and Adams 1968; Gill and Schumann 1974; Gill and Clarke 1974; Clarke 1977; Brink 1982; Freeland et al. 1986; see summary

in Brink 1991), but advancements in furthering our understanding of this coastal phenomena have largely stagnated since then. During this time, several analytical studies examined the role of more complex, but highly idealized shelf geometries on CTW scattering (Chao et al., 1979; Wang, 1980; Huthnance et al., 1986). A more robust modeling study including the





effect of stratification replicated and built upon the findings of these analytical studies (Wilkin and Chapman, 1990). However, scattering of CTWs remains a relatively large knowledge gap as much is still unknown.

CTW scattering is a rather complex process that involves redistribution of energy from one mode into another (typically higher order) mode. Interference between incident and refracted or reflected waves modifies the expected cross-shelf modal structures and propagation properties. Difficulty in separating individual modes of observed CTWs in addition to the unknown structure and energy of the incoming or incident wave has necessitated either analytical or modeling studies to investigate this phenomenon. Scattering is largely induced by non-gradual changes in coastline and bathymetry. Idealized numerical experi-

ments have an advantage over observational studies by being able to isolate individual features such as narrowing or broadening shelves, canyons, and ridges to understand their effects.

    Most continental shelves are not this simple so agreement between observed and theoretical CTWs begins to deviate once the assumptions of a straight coastline with shelf similar bathymetry are violated (Chapman, 1987; Brunner et al., 2019). The Mid-Atlantic Bight (MAB) region on the east coast of the United States has a particularly complicated continental shelf with

multiple large turns in the coastline, rapid shoaling of the bathymetry in Georges Bank, a cross-shelf canyon in Hudson Shelf Valley, narrowing of the coastline at North Carolina, and multiple estuaries (Long Island Sound, Hudson River, Delaware Bay, and Chesapeake Bay). Accordingly, there have been very few studies on CTWs in this complex region (Wang, 1979; Ou et al., 1981; Schulz et al., 2012; Brunner et al., 2019).

    We recently compared CTW velocity and pressure observations from a cross section of moorings across the North Carolina

shelf to theoretical CTW modes at that same cross section (Brunner et al., 2019). Poor agreement between observations and theory was found and it was hypothesized that this was likely due to scattering induced by the narrowing shelf. In this paper, we use high-frequency radar surface velocity data covering the MAB continental shelf to observe CTW scattering over the entire region as well as a more concentrated focus on the North Carolina shelf. A combination of two different empirical orthogonal function (EOF) analysis methods suited for decomposing propagating signals allows us to identify high scattering regions. The

observed scattering can be attributed to the various complex coastline and bathymetric features in the MAB. This is the first time that CTW scattering has been directly identified within observations. Furthermore, our relatively simple methods can be directly applied to other continental shelves around the world to identify high scattering regions.

    This paper is arranged as follows. Section 2 describes the SeaSonde high-frequency radar velocity data and EOF analysis methods used. Section 3 contains observations of CTWs along the coast and scattering results for the full MAB study region as

well as a more concentrated look at the North Carolina shelf. Section 4 discusses how the results align with previous scattering studies and the role of stratification. Finally, Sect. 5 summarizes the findings of this study.

## 2   Data and Methods

### 2.1   Data

The Mid-Atlantic Regional Association Coastal Ocean Observing System (MARACOOS) project oversees the operation of a

network of fifteen 5MHz SeaSonde stations along the US East coast (Figure 1 for locations). Rutgers University uses optimal





interpolation to pre-process the high-frequency radar (HFR) raw surface radials into Cartesian zonal and meridional velocity components (data are available at tds.marine.rutgers.edu). The operational frequency of the radar system results in a spatial resolution of 0.05º or 6 km in both the $x$- and $y$-directions and an effective depth of 2.4 m (Stewart and Joy, 1974). Hourly data are available from January 1, 2006 to present, but we focus on winter (defined as January, February, March) and summer

(June, July, August) of 2017 to observe seasonal differences.

Although the data come pre-processed, quality control of the data is still important, so we apply several cut-off criteria to ensure high quality data (Figure 1). First, we require the geometric dilution of precision (GDOP) of both velocity components to be less than 0.85. GDOP is a non-dimensional scalar (Chapman and Graber, 1997) that ranges from 0 to 1 for this data set to describe the quality of a velocity measurement, where increasing values indicate decreasing data quality. By imposing a

coupled rather than combined GDOP threshold we prevent the dilution of the quality of the data by a poor measurement from a single velocity component. Second, we eliminate data in water depth greater than 500 m as we are interested in the circulation on the shelf and data quality tends to decrease with distance offshore. Lastly, we remove data from a thin coastal region along the New Jersey, Delaware, and Maryland coastline that we have determined to have large directional uncertainty (Brunner and Lwiza, 2020).

**2.1.1  Methods**

We linearly interpolate over gaps less than six hours in length to get as complete a data set as possible without introducing significant interpolation errors. The range of periods of CTWs within the study region is identified using continuous wavelet power spectra analysis as in Camayo and Campos (2006). The data are then bandpass filtered between 3 and 12 days to isolate the identified CTW signal. Other signals may exist within this range of periods, but their contribution is small compared to the

observed CTW velocities which are $\mathcal{O}(10 \text{ cm s}^{-1})$.

Following Kaihatu et al. (1998) and Barnett (1983), we perform real vector empirical orthogonal function (R-EOF) analysis and complex empirical orthogonal function (C-EOF) analysis, respectively, on the CTW velocity data. Similar to standard EOF analysis, these methods are used to determine the variability of the CTW velocity by decomposing the time series into independent spatial modes or eigenvectors (EOFs) and their corresponding time series (principal components, PCs). The first

mode explains the highest percentage of the variance of the original data being analyzed, with each successive mode explaining less of the variance (North, 1984). We assume that each of the EOF modes correspond to a CTW mode.

Unlike standard EOF analysis, R-EOF analysis accepts the coupled vector velocity components as $[u \ v]$ and is a better method for decomposing propagating signals. R-EOF results in velocity vector maps and their corresponding PCs. This method is able to resolve correct directionality within the velocity vector maps as it maintains the vector nature of the velocity field but

provides no propagation information as it cannot determine phase. The vector maps, therefore, provide CTW flow direction and magnitude information for each mode. Similarly, C-EOF analysis is good at decomposing propagating signals but only decomposes each velocity component individually. Due to the natural propagation direction of CTWs and the orientation of the coastline, we focus on the meridional velocity component ($v$). Obtaining the Hilbert transform of the data before calculating the covariance matrix results in phase maps that provide propagation information but are unable to provide vector information.





Merrifield and Guza (1990) caution against using C-EOF on CTWs, however the data analyzed here meets their requirements
      for use.

      We, therefore, use a combination of C-EOF and R-EOF results – vector velocity maps and vector PCs from R-EOF and phase
      spatial modes and phase PCs from C-EOF. The methods provide different, but complimentary information about the CTW
      modes. The phase maps provide propagation information while the vector maps provide CTW flow direction and magnitude
information for each mode. We further assume that each of the modes from the two different methods are approximately the
      same, i.e. mode 1 from C-EOF corresponds to mode 1 from R-EOF, etc. The percentage of the variance that each mode explains
      as well as the spatial correlation help validate this assumption.

## 3    Results

### 3.1    Observations of CTWs

After using wavelet analysis to identify the range of CTW periods and bandpass filtering to isolate the identified signal, we
      first plot both the zonal and meridional CTW surface velocity components at select points along the coastline to confirm
      the presence of CTWs and observe their propagation (Figure 2). The filtered velocity signals are consistent with previously
      observed CTWs (Brunner et al., 2019) although the signal is carried in both velocity components rather than just the meridional
      component due to the coastline orientation. Identified surface velocities typically fall between -35 to 20 cm s$^{-1}$, with a negative
sign indicating typical, southward propagation. Velocity magnitudes are reduced in summer compared to winter, also consistent
      with previous observations.

      The phase speed is too fast to reasonably estimate from hourly observations; at such coarse time resolution, the signal
      appears to be largely in phase or slightly shifted right from one location to the next. It is, however, apparent that the signal
      transforms or becomes modified as it travels along the coast, likely due to changing bathymetry and scattering. Sometimes a
large signal becomes damped as it propagates (e.g., mid-March or end of July, Figure 2) and other times it becomes enlarged
      (e.g., mid-January).

      We begin to investigate the role of scattering at these locations by decomposing the winter CTW velocity signal into its
      modes using the dominant velocity component. Standard EOF analysis (North, 1984) is conducted on the observational data
      whose grid points fall along the indicated cross-sections (Figure 2 left). For the observations, the cross-sections end where
the lines intersect with the 500 m isobath due to the applied cutoff criteria. These modes are compared to theoretical EOF
      modes obtained for each of the cross-sections using K. Brink's dynamic mode analysis software (available at: whoi.edu; Brink
      2006). The dynamic mode analysis software solves for CTW free wave solutions to the Boussinesq, hydrostratic, linearized
      momentum equations,

$$\frac{\partial u}{\partial t} + v_0 \frac{\partial u}{\partial y} - fv = -\frac{1}{\rho_0} \left( \frac{\partial P}{\partial x} - \frac{\partial \tau^{tx}}{\partial z} \right), \qquad\qquad (1)$$






$$\frac{\partial v}{\partial t} + u\frac{\partial v_0}{\partial x} + v_0\frac{\partial v}{\partial y} + w\frac{\partial v_0}{\partial z} + fu = -\frac{1}{\rho_0}\left(\frac{\partial P}{\partial y} - \frac{\partial \tau^{ty}}{\partial z}\right), \tag{2}$$

$$\frac{\partial P}{\partial z} = -g\rho_2, \tag{3}$$

$$\frac{\partial u}{\partial x} + \frac{\partial v}{\partial y} + \frac{\partial w}{\partial z} = 0, \tag{4}$$

$$\frac{\partial \rho_2}{\partial t} + u\frac{\partial \rho_1}{\partial x} + v_0\frac{\partial \rho_2}{\partial y} + w\frac{\partial \rho_1}{\partial z} = 0, \tag{5}$$

where $x$ is the cross-shelf direction, $y$ is the alongshelf direction with positive northward, and $z$ is depth positive upward; $u(x,z,t)$, $v(y,z,t)$, and $w(x,y,z,t)$ are the velocities; $v_0(x,z)$ is the mean flow; $\rho(x,y,z,t) = \rho_0 + \rho_1(x,z) + \rho_2(x,y,z,t)$ is

the density as a combination of the mean density, $\rho_0$, background density, $\rho_1$, and wave perturbation density, $\rho_2$; $P(x,y,z,t)$ is the perturbation pressure; and $\tau^{tx}(x,z)$ and $\tau^{ty}(y,z)$ are the turbulent stresses. Additional details can be found on the software website or in Brink (2006) and Brunner et al. (2019). This formulation and others rely on an assumption of a straight coastline with shelf similar bathymetry so they are incapable of capturing CTW scattering. Therefore, differences between observed and theoretical modes are indications of possible scattering as the model is unable to capture this behavior.

Generally, the shape of each of the observed modes agrees with the modeled modes but overall agreement is very poor (Figure 3). Agreement is better for MA, NJ, and DE, which show reasonable mode 1 agreement while secondary peaks in mode 2 and 3 are shifted toward the shelfbreak. All of these locations are on relatively straight coastlines and also have the greatest percentage of the variance explained by mode 1 (Table 1), indicating these may not be high scattering regions. Secondary peaks in mode 2 and 3 are shifted toward the coast for NC and mode 1 peaks offshore rather than at the coast, as it

does for HC and DE as well. Percentage of the variance explained indicates that a great deal of energy has been shifted from mode 1 to mode 2, likely due to scattering. Finally, observed HC modes are very complex and show very little agreement with the modeled modes. Scattering is expected in this highly complex region (Zhang and Lentz, 2017, 2018).

## 3.2 Full shelf scattering results

Having confirmed the presence of CTWs and identified some possible regions of scattering, we utilize the full shelf R-EOF

and C-EOF results to properly identify regions of CTW scattering. Amplification or reduction in vector magnitude, rotation of vectors away from an alongshelf direction, and large, sudden jumps in phase are all indications of scattering (Wang, 1980). We focus first on the winter results (Figure 4), which we assume have a well-mixed water column. Vector and phase PCs are largely in phase and have a similar pattern for each mode despite obvious magnitude differences, allowing us to interpret the selected R-EOF and C-EOF results together. The first three modes of R-EOF explain 45.5%, 17.9%, and 7.9% of the variance





and requires a cumulative 17 modes to reach the 90% variance threshold. PCs are similar to the zonal and meridional wind

components (Figure 4g), indicating that these are wind-forced features that we are observing. The first three modes of C-EOF

explain 51.7%, 14.5%, and 8.7% of the variance.

Mode 1 flow is unidirectional in the alongshelf direction over the vast majority of the shelf (Figure 4a). Flow north of

Georges Bank (GB) in the Gulf of Maine (GOM) is also unidirectional and alongshelf, but pointing in the opposite direction.

This area is also approximately 180$^{\circ}$ out of phase with the rest of the shelf, which is entirely in phase (Figure 4d). Vector

magnitude in the GOM is reduced compared to the rest of the shelf, but is amplified after crossing into GB. These factors

indicate that this is a scattering region for mode 1 CTWs. Vector magnitude is also reduced within the Hudson shelf valley

(HSV) and along the North Carolina (NC) coast. Although there is very little change in phase associated with this reduction in

amplitude, this is also indicative of some mode 1 scattering and absorption.

Mode 2 flow is more complicated. Flow is no longer unidirectional but cross-shelf or perpendicular to bathymetric contours

in several areas (Figure 4b). These areas include GB, HSV and the shelf just south of it, and near Delaware Bay. Regions

of cross-shelf flow are associated with 180$^{\circ}$ phase jumps (Figure 4e). Continuing south of the Delaware Bay region, the NC

shelf is also part of the scattered phase region although flow is not cross-shelf. Flow along the NC shelf is actually alongshelf

but in the opposite direction of typical flow, as seen on the Southern New England (SNE) and New Jersey (NJ) shelf. This is

indicative of either backscattering or reflection of the mode 2 CTW.

In mode 3, the GOM/GB area again shows a divergence of flow direction and amplification of vector magnitude (Figure 4c)

as well as a change in phase (Figure 4f). Amplitude is reduced over a greater area of the HSV and the flow field is convergent.

Amplitude is also reduced along the NC coast. GOM/GB, HSV, and NC are common regions that show scattering in all three

modes, suggesting that these are high scattering regions. Rotation and slight funneling near Delaware Bay in mode 2 suggests

that estuaries, which are an interruption to the waveguide, may also induce scattering.

It has been proposed that stratification increases scattering into a higher proportion of higher order modes (Wilkin and

Chapman, 1990), so we compare the well-mixed winter case to a stratified summer case (Figure 5). The first three modes of

R-EOF explain 41.6%, 12%, and 8% of the variance and requires 30 modes to reach the 90% variance while the first three

modes of C-EOF explain 45%, 15.6%, and 9% of the variance. Percentage of the variance explained by the first three modes has

decreased and the number of modes required to reach the 90% variance threshold has increased. This indicates that all modes,

not just mode 1, are scattering into progressively higher modes in a cascade (Chao et al., 1979) with increased stratification.

Mode 1 is very similar to the winter case with flow unidirectional in the alongshelf direction. Flow north of GB is again

pointing in the opposite direction (Figure 5a), although this area has decreased. Absence of data in this region does not allow

us to determine if there is a phase jump (Figure 5d). Amplitude is not reduced over the HSV, suggesting stratification prevents

the CTW from feeling the effect of the canyon as much. Reduced amplitude near Long Island Sound and Chesapeake Bay also

suggests that increased freshwater inflow from the estuaries may also induce scattering.

Mode 2 and 3 vector maps also look quite similar to the winter case. In mode 2 (Figure 5b), flow is again cross-shelf in GB,

HSV, and Delaware Bay although there is more apparent funneling toward the Delaware and Chesapeake Bay estuary mouths.

The reflection near NC is reduced in mode 2 but greatly amplified in mode 3 (Figure 5c), suggesting that this interaction may





have been scattered into a higher order mode by stratification. Notably, the influence of HSV is absent where it was so prevalent for the winter case. This is also true of the phase maps (Figure 5e and f), which are more complex than for the winter case although phase jumps of nearly 180º are still present in these common areas.

We also evaluate the seasonal CTW contribution to the total kinetic energy over the continental shelf. Following Takikawa et al. (2003), the ratio of CTW to total kinetic energy is calculated as

$$R_E = \frac{\bar{u_c^2} + \bar{v_c^2}}{\bar{u^2} + \bar{v^2}} \times 100 \tag{6}$$

where $u_c$ and $v_c$ are the seasonal CTW velocities, $u$ and $v$ are the annual mean currents, and the overbar indicates a temporal average. The average summer contribution of CTWs to the total kinetic energy is 23.3%, while the winter contribution is 30.3%. This supports the seasonality of the CTW velocity magnitude that we observed in Brunner et al. (2019). The average CTW winter contribution is also approximately the same as the tidal contribution (Brunner and Lwiza, 2020), indicating that

CTWs are an equally important component of the overall shelf circulation and together they contribute ∼60% of the total kinetic energy. Spatial patterns indicate that the winter contribution is higher along the SNE, on either side of HSV, and along the NC coast (Figure 6).

### 3.3 North Carolina shelf scattering results

Next, we repeat the above procedure for the high scattering region of the NC shelf identified above and in our previous study

(Brunner et al., 2019). By de-coupling it from the remainder of the shelf, we can determine if there are separate or additional scattering processes occurring there and also observe smaller scale scattering features than we can for the entire study region.

In winter, the first three modes of R-EOF explain 44.4%, 15.5%, and 8.1% of the variance and requires a cumulative 13 modes to reach the 90% variance threshold and the first three modes of C-EOF explain 47%, 17.5%, and 11.1% of the variance. In summer, the first three modes of R-EOF explain 48.4%, 16.2%, and 7% of the variance and requires a cumulative 11 modes

to reach the 90% variance threshold and the first three modes of C-EOF explain 48.4%, 16.8%, and 10.1% of the variance. Unlike the entire study region results, we do not see a decrease in the mode 1 variance explained from winter to summer and the number of modes required to reach the 90% threshold does not increase.

Vector and phase maps are consistent with the entire study region results but show higher resolution features (Figures 7, 8). Amplification of the vector magnitude near the southern extent of the domain in all three modes of both seasons is consistent

with shelf narrowing. Mode 1 flow is unidirectional with reduction of the amplitude near the Chesapeake Bay mouth and amplification near Cape Hatteras (Figures 7a, 8a) and has a consistent phase (Figures 7d, 8d). Divergence of the flow at the Chesapeake Bay mouth and cross-shelf flow near Cape Hatteras is stronger in summer than winter. These areas are associated with 180º phase jumps. Lastly, the mode 3 vector maps show the reflected or backscattered CTW (Figures 7c, 8c) which is much stronger in summer and likewise associated with phase changes (Figure 8f).

We also examine the energy density, $(u^2 + v^2)/2H$, within this region using the summer R-EOF vector map results for $u$ and $v$ and the water depth for $H$ to determine how the scattering process modifies the CTW energy transmission. This requires





us to assume that the velocities are consistent vertically. We have previously shown that CTWs are barotropic in this region even when the water column in stratified (Brunner et al., 2019), so this is a valid assumption.

We do not know the maximum value of the incident wave to normalize the energy density per Wang (1980). Therefore, we
are forced to use some other value. The calculated energy density distribution is strongly right skewed and its maximum value is an outlier. Hence we normalize by the value corresponding to the third standard deviation from the average, $\mu + 3\sigma$ (i.e., representing a probability of 0.003 of exceeding that value). Since we are unable to normalize by the incident energy density, the normalized values do not have the physical meaning of indicating whether scattering is increasing or decreasing energy density transmission relative to the incident wave. Our analysis is, therefore, more qualitative but still highlights regions of
energy modification.

We find that there are three primary areas where energy is increased (Figure 9). These areas are the Chesapeake Bay mouth, the coast, and the southern extent of the domain by Cape Hatteras. These areas correspond to scattering identified using the vector and phase maps (Figures 7 and 8), thus we can attribute this increased energy to scattering. The shelf narrowing to a minimum shelf width around Cape Hatteras and the Gulf Stream, which opposes typical CTW propagation direction, clearly
have a large effect on scattering and energy transmission as its influence is seen to some extent in all three modes. The Chesapeake Bay mouth and estuarine inflow appears to have the greatest effect on mode 2, while the narrowing shelf width influence of the coast has the greatest effect on both modes 1 and 2.

## 4 Discussion

Guided by scattering principles determined through theoretical (Wang, 1980) and modeling work (Wilkin and Chapman, 1990),
we are able to unequivocally identify CTW mode scattering within observations for the first time. We previously theorized that scattering explains the poor agreement between observed and theoretical CTW modes off the coast of NC (Brunner et al., 2019) but did not have the necessary framework to demonstrate that scattering was occurring. We can now show that a combination of R-EOF and C-EOF analyses can be effectively utilized to identify CTW scattering within both full shelf and regional scales. Reduction/amplification of vector magnitude and/or rotation of vectors into the cross-shelf direction on the R-EOF vector maps
and large, sudden jumps in phase on the C-EOF phase maps are indications of scattering.

Using these guidelines, we identify the GOM/GB transition zone, HSV, Delaware Bay mouth, Chesapeake Bay mouth, and NC shelf as high scattering regions. Bathymetry in the GOM/GB transition zone is very complex. The shelf is extremely thin along the Cape Cod coast before broadening into shallow GB and the SNE shelf. Generally, broadening of a shelf induces scattering into higher order modes (Wilkin and Chapman, 1990) and reduction of the velocity amplitude (Wang, 1980) although
the large bathymetric step from GOM to GB which increases velocity amplitude must also be considered. Accordingly, we observe divergence of the flow at the GOM/GB transition zone as the incoming wave is reflected, amplification of the magnitude in shallower GB, and increased cross-shelf flow due to scattering into mode 2.

Canyons, and HSV in particular, have garnered previous attention (Wang, 1980; Zhang and Lentz, 2017, 2018). Only part of the incoming wave is transmitted as a portion of the wave is reflected as short wave. This short wave reflection is largely





cross-shore and confined to the canyon region due to friction. The transmitted wave is amplified due to convergence towards
the tip of the canyon (Wang, 1980). However, orientation of the canyon with respect to propagation and wind forcing direction
results in asymmetrical flow with onshore flow developing either weakly over the canyon or stronger upstream of the canyon
(Zhang and Lentz, 2017). While we do see onshore flow associated with HSV, the overall pattern differs from this expected
behavior. This is likely because the location of HSV is also associated with a nearly 90° turn in the coastline which will induce
its own scattering, while these studies are for canyons on an idealized, straight shelf.

The effect of narrowing shelves similar to the NC shelf has also been previously examined (Wang, 1980; Wilkin and Chap-
man, 1990). Decreasing phase speed with decreasing shelf width amplifies the wave and the percentage of the incoming energy
that gets reflected is proportional to the geometry of the shelf (Wang, 1980). Furthermore, the effect of scattering within this
region can also be found upstream of the scattering region, particularly in the cross-shelf component (Wilkin and Chapman,
1990). We observe amplification, reflection, and cross-shelf scattering on the NC shelf, as expected. We also find an increase
in the energy density associated with these indicators of scattering. The proximity of the Chesapeake Bay mouth complicates
scattering just upstream of the NC scattering region so it is not clear whether the cross-shelf flow is due to the upstream influ-
ence of the narrowing shelf or due to the estuary. The effect of estuaries on CTWs is not well understood, although we observe
scattering at both Chesapeake Bay and Delaware Bay. Flow scattered into the cross-shelf direction is not directly perpendicular
to bathymetry but tends to funnel towards the estuaries. This is also a behavior we have previously observed, but were unable
to attribute directly to scattering (Brunner and Lwiza, 2019).

Most previous studies ignore the role of stratification due to the complexity of the problem (Chao et al., 1979; Wang, 1980).
The use of computational models allows the effect of stratification to be investigated rather simply. Wilkin and Chapman
(1990) found that scattering by irregular topography is increased by stratification, while Zhang and Lentz (2018) found that
asymmetrical flow generated by CTWs over a shallow shelf valley such as HSV is insensitive to stratification. Theoretical
CTW mode results from the Brink (2006) analytical model for the NC shelf also suggest that seasonal stratification is not an
important factor in that high scattering region either (Brunner et al., 2019).

Our results are able to rectify these contrasting views and fall more in line with Wilkin and Chapman (1990). Stratification
does have an effect on the modes and has a greater effect on higher order modes, although mode 1 is also impacted. The
number of modes required to reach the 90% variance explained threshold doubles from winter to summer for the entire shelf
results. This is likely due to a coupling of stratification increasing scattering to higher order modes (Wilkin and Chapman,
1990) and a net cascading process induced by the complex bathymetry (Chao et al., 1979). Interruption of the waveguide and
increased freshwater input from estuaries also have a larger impact in summer, indicated by changes in phase as well as more
prominent rotation and funneling toward the estuary mouths. Interestingly, the scattering role of HSV is diminished in summer
and nearly disappears from mode 1. This directly contradicts what Zhang and Lentz (2018) found and suggests the need for a
more concentrated study in that area.

Our simple methodology can be easily applied to CTW observations from other complex coastlines around the world. Most
observational studies have been conducted on relatively straight coastlines with smooth bathymetry, but several other studies
have also found deviations between theory and observations. Battisti and Hickey (1984) and Chapman (1987) attributed these





differences along the California and Oregon coastlines to error in the model wind forcing. However, we find it more likely that these differences arise due to scattering caused by complexities in the coastline near the Channel Islands and Point Conception, the San Francisco Bay mouth, Point Reyes, and Salish Sea. A modeling study of the shelf narrowing at Lofoten, Norway did not include the effects of scattering (Drivdal et al., 2016), but observations should show the effects of a narrowing and widening shelf. Observations of CTW sea surface height from the coast of Australia did not show large changes in phase or

amplitude associated with topographic features such as shelf narrowing at Portland or complexity at Bass Straight (Woodham et al., 2013), suggesting that CTW velocity may be more sensitive to scattering than sea surface height.

## 5   Conclusions

SeaSonde HFR surface velocity data allow us to take a detailed look at CTWs propagating on the MAB continental shelf. Poor agreement between theoretical and observed CTW modes (Figure 3; Brunner et al. 2019) as well as the complex coastline

and bathymetry in the MAB suggest that CTW scattering is prominent throughout this region. We use R-EOF and C-EOF to decompose the CTW velocity into its modes. Results from these two analysis methods allow us to identify regions of CTW scattering within the MAB. This is the first time that scattering has been identified using observations rather than analytical or computational models. Changes in vector magnitude and rotation of vectors into the cross-shelf direction on the R-EOF vector maps and phase jumps on C-EOF phase maps are indications of scattering. We identify the GOM/GB transition zone, HSV,

Delaware Bay, Chesapeake Bay, and NC shelf as high scattering regions using these guidelines. CTW scattering is, therefore, common throughout much of the MAB continental shelf.

Mode 1 is largely unidirectional and typically accounts for approximately 50% of the variance. This percentage decreases in summer when stratification increases scattering into a greater percentage of higher order modes. Furthermore, a cascade of scattering into progressively higher modes requires collectively many more modes to describe 90% of the total variance.

Stratification also greatly increases the scattering influence of estuaries, which has not been previously examined. This is one aspect of scattering that is not particularly well understood and requires additional attention. The scattering influence of HSV also does not follow what we expect based on previous studies (Wang, 1980; Zhang and Lentz, 2017, 2018), perhaps due to its location within a large coastline turn, and should be studied further. However, the amplification, reflection, and cross-shelf scattering of the narrowing shelf off the coast of NC does align with expectations and explains poor agreement with theory in

our previous study (Brunner et al., 2019). Our methodology can be applied to other coastlines globally to identify additional high scattering regions and gain a better understanding of the CTW scattering process. Possible future work involves the set-up of a hydrodynamic model such as ROMS or HYCOM to compare to observations and quantify the amount of scattering.

*Code availability.*   K. Brink's dynamic mode analysis software is available at https://www.whoi.edu/page.do?pid=23361 (Brink, 2006).



*Data availability.* HFR surface velocity data from Rutgers University and the MARACOOS project are available at http://tds.marine.rutgers.

edu/thredds/dodsC/cool/codar/totals/5Mhz_6km_realtime_fmrc/Maracoos_5MHz_6km_Totals-FMRC_best.ncd.html.

*Author contributions.* K. Brunner and K. M. M. Lwiza both helped design the study while K. Brunner performed the analysis. K. Brunner also prepared the manuscript with contributions from K. M. M. Lwiza.

*Competing interests.* No competing interests are present.

*Acknowledgements.* We would like to thank Hugh Roarty (Rutgers University) for helping us obtain the data and answering our questions

regarding pre-processing and quality control. We would also like to thank Dong-Ping Wang (Stony Brook University) for discussions about CTW propagation and scattering, particularly in the complex MAB.





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





**Figure 1.** HFR spatial coverage for 2017. Dashed black lines indicate full coverage and solid black lines indicate controlled coverage after applying the cutoff criteria. Diamonds indicate locations of SeaSonde stations. Light grey lines are bathymetric contours at $z = 20$, 40, 60, 80, 200, 500, 1000, 2000, 3000, and 4000 m depth. Blue lines outline the shelf regions from north to south which are Gulf of Maine (GOM), Georges Bank (GB), Southern New England (SNE) shelf, Mid-Atlantic Bight (MAB) shelf, and North Carolina (NC) shelf. Additional referenced locations indicated in red from north to south are Nantucket Island (NI), Long Island Sound (LIS), Hudson River (HR), Hudson Shelf Valley (HSV), Delaware Bay (DB), Chesapeake Bay (CB), and Cape Hatteras (CH).

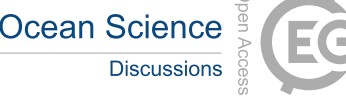



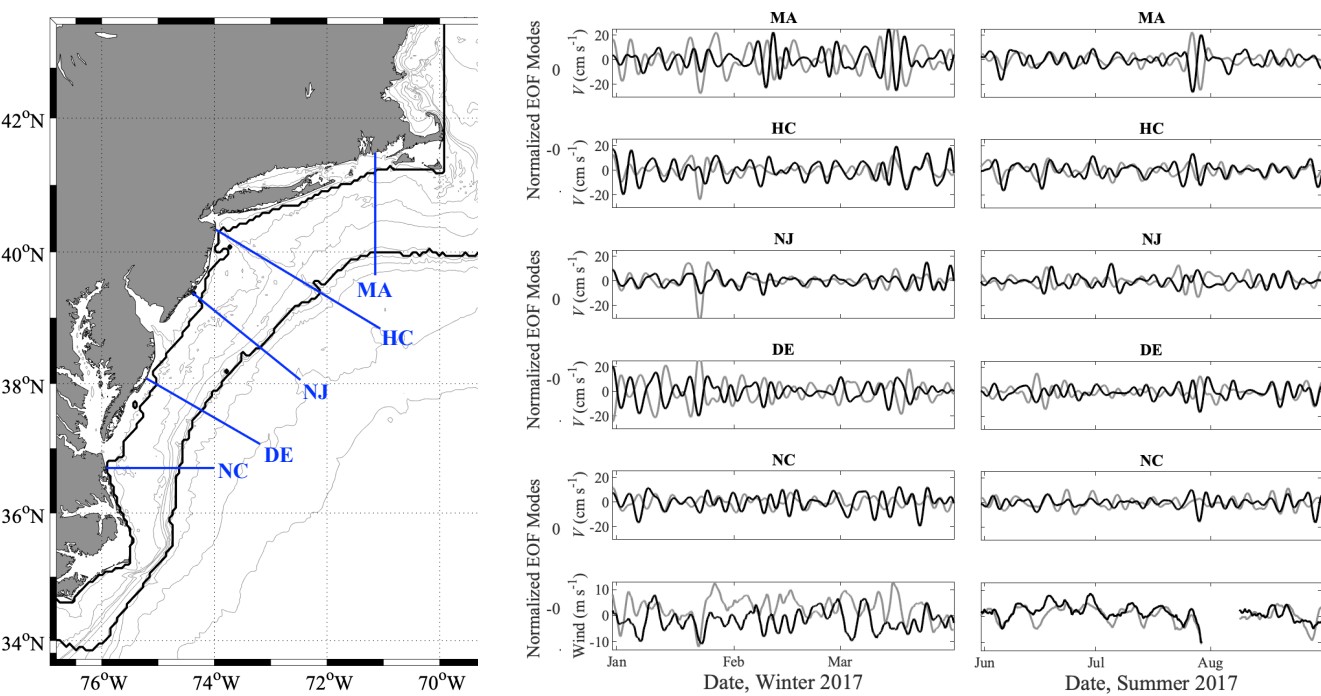

**Figure 2.** (Left) Shelf cross sections used for CTW mode analysis (blue lines). Solid black lines indicate controlled HFR coverage and light grey lines are bathymetric contours, as in Figure 1. (Center, Right) CTW velocity amplitude (cm s$^{-1}$) for the five US east coast locations corresponding to the point closest to shore from the cross-sections on the left. Displayed are filtered values for the zonal (grey lines) and meridional (back lines) surface velocity in winter (center column) and summer (right column). Zonal (grey line) and meridional (black line) wind components from buoy 44025 are shown in the bottom panel.

**Figure 3.** Winter EOFs. Displayed are the first three modes (mode 1, black lines; mode 2, red lines; and mode 3, blue lines) of normalized surface velocity patterns from the Brink model (thin, dashed lines) and EOF analysis (thick, solid lines).





**Table 1.** Percentage (%) of the variance explained by each of the EOF modes.

| Mode | MA | HC | NJ | DE | NC |
|------|------|------|------|------|------|
| 1 | 85.4 | 77.6 | 88.2 | 85.3 | 51.5 |
| 2 | 7.7 | 11.0 | 5.0 | 10.3 | 20.7 |
| 3 | 2.3 | 4.8 | 2.4 | 1.9 | 8.6 |





**Figure 4.** (a-c) R-EOF vector maps and (d-f) C-EOF phase maps for mode 1 (left column), mode 2 (middle column), and mode 3 (right column), and (g) the corresponding PCs (R-EOF red lines, C-EOF blue lines) of CTWs for winter 2017, entire study region. Variance explained by each mode is indicated. Zonal (grey line) and meridional (black line) wind components from buoy 44025 are shown in the bottom panel of (g). Note that vectors are plotted every 4 grid points for visibility.







**Figure 5.** (a-c) R-EOF vector maps and (d-f) C-EOF phase maps for mode 1 (left column), mode 2 (middle column), and mode 3 (right column), and (g) the corresponding PCs (R-EOF red lines, C-EOF blue lines) of CTWs for summer 2017, entire study region. Variance explained by each mode is indicated. Zonal (grey line) and meridional (black line) wind components from buoy 44025 are shown in the bottom panel of (g). Note that vectors are plotted every 4 grid points for visibility.





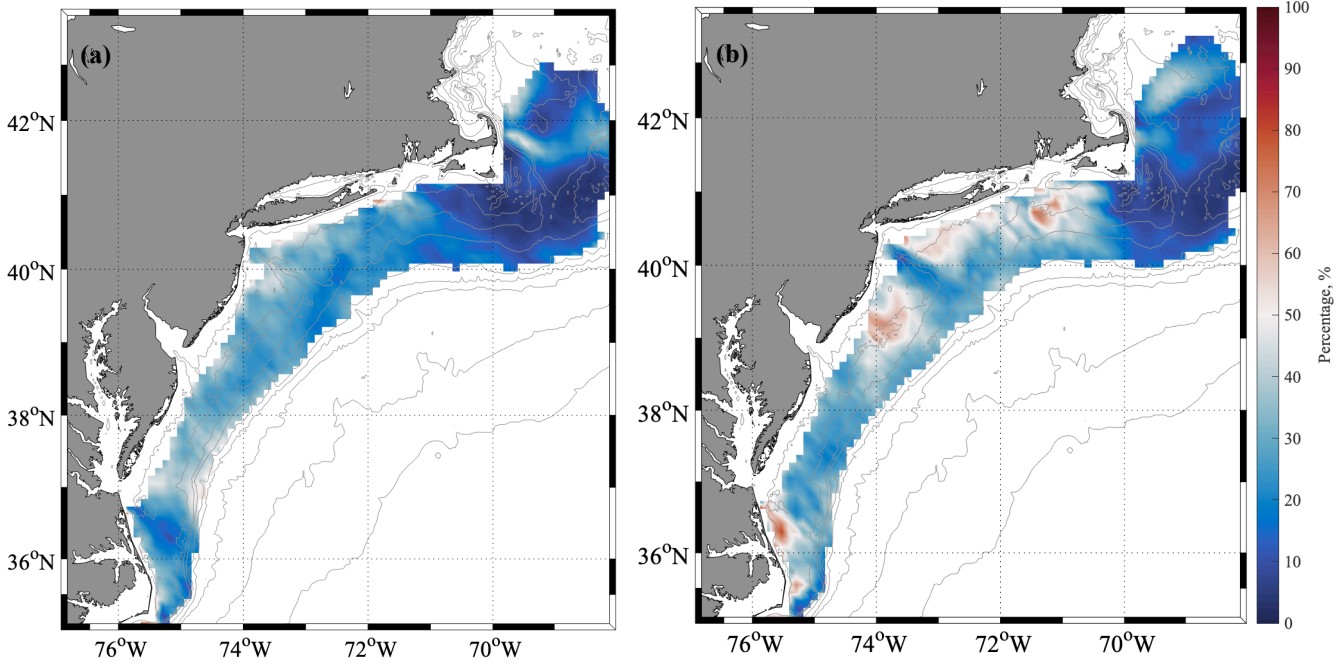

**Figure 6.** The ratio of (a) summer and (b) winter CTW current kinetic energy to annual mean current kinetic energy as a percentage (%).





**Figure 7.** (a-c) R-EOF vector maps and (d-f) C-EOF phase maps for mode 1 (left column), mode 2 (middle column), and mode 3 (right column), and (g) the corresponding PCs (R-EOF red lines, C-EOF blue lines) of CTWs for winter 2017, NC shelf. Variance explained by each mode is indicated. Zonal (grey line) and meridional (black line) wind components from buoy 44025 are shown in the bottom panel of (g). Note that vectors are plotted every two grid points for visibility.





**Figure 8.** (a-c) R-EOF vector maps and (d-f) C-EOF phase maps for mode 1 (left column), mode 2 (middle column), and mode 3 (right column), and (g) the corresponding PCs (R-EOF red lines, C-EOF blue lines) of CTWs for summer 2017, NC shelf. Variance explained by each mode is indicated. Zonal (grey line) and meridional (black line) wind components from buoy 44025 are shown in the bottom panel of (g). Note that vectors are plotted every two grid points for visibility.



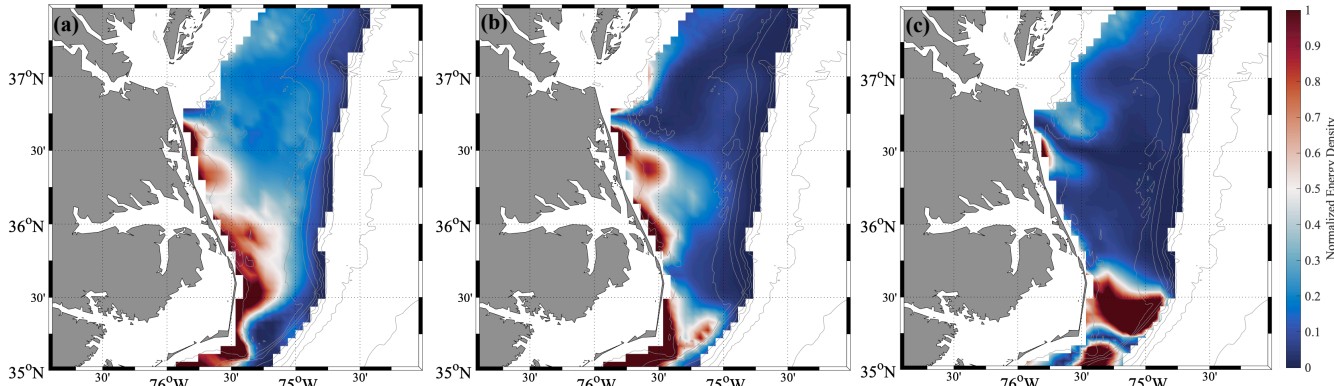

**Figure 9.** (a) Mode 1, (b) 2, and (c) 3 normalized energy density for summer 2017, NC shelf.