# Peer review of "Evidence of coastal trapped wave scattering using high-frequency radar data in the Mid-Atlantic Bight"

_Ocean Science, 2020_

## Referee Comment (RC1) · Anonymous Referee #1 · 30 Jul 2020

**General comments**:

- This article proposes an analysis of HF-radar observations of surface currents over the Mid-Atlantic Bight in terms of coastal trapped waves dynamics, using EOF methods, with a strong emphasis on the "scattering" of low-order CTW modes into higher-order modes by topographic/bathymetric features of the domain.

- The article is well-written, clear, and the figures are of good quality.

- The survey of the existing literature is well-written and fair.

[Figure]

- It is a follow-up on a previous study published by the same authors in Brunner et al, J. Phys. Oceanogr., (2019), based on a different dataset (in-situ mooring observations from the OMP field program).

- As an observational oceanographer, I am impressed by the courage and determination the authors have shown in getting to grips with these two very different and I imagine bulky datasets.

- My concern is that the analysis in both papers appears to me as extremely biased, CTWs and their dynamics being proposed as the sole and only possible explanation for the observed surface currents features, the -very large- discrepancies between observations and this theoretical framework being systematically ascribed to "scattering" processes of the CTWs. In all domains of physics I know of, wave propagation models are only considered useful in situations where the order-0 phenomenon is propagation and other mechanisms such as forcing, dissipation or scattering in response to environmental heterogeneities can be relegated to higher orders. I have not seen evidence of wave propagation as a dominating phenomenon in either of these articles (except for the black line in Fig11b of the JPO paper), and I have not found a statement of the existence of CTWs as a leading-order process in the MAB in the literature cited by the authors. I am thus under the impression that the surface currents dynamics in the MAB should not be described as CTWs dynamics, perturbed by the geometry of the domain (which is the analysis angle advocated by the authors), but by a range of processes, among which CTWs may or may be. I thus consider that the authors' main claim that they have observed "scattering" of CTWs is not substantiated.

- In fact, rather than an attempt at assessing the impact of CTWs on the MAB dynamics, which does not seem very promising in the view of the paper as it stands, I think an unbiased analysis of the dataset aimed first at extracting its dominant features and identifying the dominant physical processes governing MAB dynam-

ics would be more interesting. This however appears like a completely different paper, and exceeds even the scope of major revisions. The current article should thus probably rejected, or put on hold by the authors until sufficient evidence of its relevance can be added to it.

**Specific comments:**

- an alternative to wave-mediated propagation of external forcing that could explain a coherent response at the MAB scale could be local responses to meteorological forcing coherent over the MAB scale. Has the MAB been analyzed along these lines?

- I am rather skeptical regarding the use of C-EOFs. I admit extracting propagating signals from a dataset is a difficult task, but I'm not sure C-EOFs is a satisfactory tool for that. I do not know if a R-EOF method considering as variables the band-pass filtered [u v du/dt dv/dt] has been proposed and studied in the signal processing literature. Such a method would introduce time variation in a computationally lighter way than time-lagged covariances, and preserve the mathematical structure of the problem, which is completely lost in the construction of the complex velocity signal.

**Technical corrections:**

---

## Short Comment (SC1) · 7 Aug 2020

The authors discuss a question that has hitherto proved difficult; scattering seems to be sensitive to topography in ways that are hard to predict from the form of topographic variation. The following comments are intended to help the clarity of the manuscript.

In the Abstract (and elsewhere in various forms) "A combination of velocity vector maps from real vector empirical orthogonal function (REOF) analysis and phase maps from complex empirical orthogonal function (C-EOF) analysis allow the identification of CTW scattering by assuming each EOF mode corresponds to a CTW mode". I agree with "identification of . . . scattering" (Reviewer 1 has raised the question whether it is

mainly CTWs). However, I don't think the authors need to "assume". Rather, they make a comparison between the EOF modes and CTW mode structures to help the "identification". EOF modes emphasise coherent aspects of the flow; where scattering occurs, EOF mode 1 (and possibly higher modes according to context) can be expected to include both the incident and scattered forms.

Towards the end of the Abstract "The simple methodology used here can be applied to observations of CTWs on other coastlines around the world." True, except that the methodology does depend on having a good coverage of the current field over the continental shelf (and slope to some extent), probably from HF radar or a model. The authors were "lucky" in having such extensive HF radar coverage. Many places around the world would not easily obtain such data coverage.

Before and after equation (6). ". . the ratio of CTW to total kinetic energy is calculated as . . . where uc and vc are the seasonal CTW velocities, u and v are the annual mean currents. . ." I think this cannot be correct. Much kinetic energy would surely be "lost" if "u" and "v" were averaged over a year before being squared. Surely the annual mean must be of the squared velocity.

Line 215. ". . energy density, $(u^2 + v^2)/2H$ . .". This is ambiguous with respect to whether "H" is a multiplier or divisor. From figures 8 and 9 I guess the latter, but then this is not "energy density". No factor H would be for energy density per volume, a multiplier H would be for energy density per area.

A personal gripe: the use of "Mid-Atlantic". It is very unfortunate that "Middle Atlantic Bight", meaning middle of the Atlantic Bight, has generally morphed to "Mid-Atlantic Bight" meaning a bight in the middle of the Atlantic – which is nonsense. Now it seems we have in MARACOOS (beginning of section 2.1) a coastal ocean in Mid-Atlantic (no mention of "bight"). I suppose the authors cannot do anything about this.

---

## Short Comment (SC2) · 21 Aug 2020

1. When the reviewer states that, 'CTWs and their dynamics being proposed as the sole and only possible explanation for the observed surface currents features, the -very large- discrepancies between observations and this theoretical framework being systematically ascribed to "scattering" processes of the CTWs.' They are overlooking the fact that in the methods section we describe how we bandpassed the current signal between 3 and 12 days. Therefore, we are excluding all the small scale (temporal) processes including signals caused by some of the short synoptic scale weather forcing. The next statement regarding the propagation of order-0, with the scattered signal

translating into higher orders is in fact demonstrated by showing that energy scattered from mode 1 into modes 2 and 3. Also, not finding other papers describing this phenomenon should not be the basis to refute its (scattering) existence. It must be noted observation of coastal trapped waves of the type we describe have only been made possible by the advent of HF radar. Therefore, it should not be surprising that this is the first paper that provides such evidence (see a comment by John Huthnance, one of the people who pioneered observations and theory of CTW).

2. The second issue is on the reviewer being skeptical of the use of the C-EOF method in extracting propagating signals. We kindly request the author to read the literature using this method and some of the references we provide in the manuscript. It is also used extensively in Atmospheric Physics. In addition, the reviewer can easily use matlab or python and create a propagating field with some noise and try out the C-EOF and compare it to what he proposes [u du/dt] ( I have ignored the v component for simplicity). I tried the comparison, and C-EOF reproduces the phase very well, except where the signal is almost equal to the error, whereas the proposed method gets the general phase pattern, but broadens the peaks. In any case, the state of the reviewer being unfamiliar with a method should not be a criterion to disqualify the method or the manuscript.
* * *
[Figure]

**Fig. 1.** Complex EOF of simple propagating wave signal from left to right with a linearly decreasing amplitude offshore

[Figure]

**Fig. 2.** Real EOF with [u du/dt] as suggested by Reviewer#1 everything else similar to Fig. 1

---

## Short Comment (SC3) · 28 Aug 2020

Dear Dr. Huthnance it is an honor for you to comment on our paper. We are very grateful for your comments. I would like to respond to your last three paragraphs.

1. The goal of calculating the ratio of CTW kinetic to the total kinetic energy was to assess the significance of CTWs. Given their nature being seasonal we computed the energy by squaring the velocities before averaging during the season when they are most prevalent, not the whole year. However, the denominator was the annual average of the squared velocities. Squaring before averaging prevents the signals canceling each other. I must admit in the manuscript as it appears on the website the overbar is

very small. We will have to find a way to make it larger.

2. With regards to wave energy density, we borrowed this idea from Dong Ping's paper of 1980. He used it in a similar fashion that is applied to wind wave energy density per area. If we take $u^2 + v^2$ to represent energy, then energy density $E = (u^2 + v^2)/AH$, where A is area and H is depth. Therefore, energy density per unit area $E' = (u^2 + v^2)/H$. Based on your comment we need to emphasize that our computation represents energy density per area.

3. A note on the personal gripe. With all due respect sir, you will find that language (or use of) is not always logical. When I first got in New York I fought against the term, but after being ignored for a long time, I realized that the locals can call it what they want. Good example is Lake Nasser for Aswan Dam, and a physical oceanography example is 'tsunami', which was borrowed from the Japanese language (meaning harbor wave as in seiche) to avoid use of the term tidal wave!
* * *

---

## Referee Comment (RC2) · Anonymous Referee #1 · 6 Sep 2020

Reply to short comment by K. Lwiza:

**Reply to point 1**:

- I am not as familiar with the phenomenology of the Mid-Atlantic Bight as the authors. For that reason, I find the position of the authors that CTWs are the only physical process left to explain the band-pass filtered velocity signal needs to be substantiated. The MAB has been the subject of very many studies, whether observational of numerical. I'm surprised the authors could not propose any reference to support their view.

[Figure]

- I'm sorry my discussion of wave phenomena (order-0, etc...) has been misleading. The point I was trying to make is that waves, basically, propagate. And that I was surprised that the authors had ascribed their failure to observe propagation in their JPO paper (I'm perfectly happy with that conclusion) to scattering, which is in my opinion an even harder process to observe unambiguously, and had put all their effort on this one hypothesis, seemingly to the exclusion of any other.

**Reply to point 2**:
I apologize for not having read carefully enough the original article. There are many methods termed "EOF" around. When I had tried to understand which version of which the authors were using, I had read the Kaihatu et al (1998) paper, which mentioned a complex-vector EOF method with which I completely disagreed. I am a lot happier with the C-EOF method described in Barnett et al 1983, which I found the authors have used. (I'm a bit curious about the practical implementation of the Hilbert transform, of which I've personally always preferred to stay clear). Also, I think this method could be applied jointly to the U and V fields, removing the need for the R-EOF method.

**Point 3:**
One point I am also not happy with is the fact that even the sensible incarnations of the EOF methods are limited to extracting eigenvectors which are orthogonal with respect to the Euclidean scalar product, while it is a classical result that the CSW modes are not (see the paper by Huthnance 1975). This is something I did not mention in my first review, as I felt I needed to expand on it quite a bit to make the review constructive, and I was too unhappy already with the C-EOF method I thought the authors had used. Basically, the method used by the authors seeks only eigenmodes which possess a property the "true" eigenmodes don't possess. The first eigenmode structure may be correctly obtained, but the subsequent eigenmodes *have to* point in the wrong direction, as the method wrongly requires them to point orthogonally to the previous modes. I am thus skeptical that the variance in the subsequent modes, which are not correctly determined, ends up there because of scattering of the first mode.

I have been uneasy about this particular point for many years, ever since I read the Huthnance 1975 paper, in fact. Orthogonality is always linked to a particular scalar product, and I felt that it should have been possible to define a scalar product for which at least some of the modes discussed in this paper would be orthogonal. I'm attaching a text in which I explain how I think this can be done, and how the C-EOF method could be modified to make the empirical eigenvectors orthogonal with respect to the correct scalar product. This analysis is restricted to the barotropic CSW case, but the authors only have access to surface currents anyway, and they will probably agree that the text is complicated enough... I imagine J. M. Huthnance, who is editor for this journal, and has already posted a comment, will have opinions on this. I agree I'm not happy with the boundary condition on the shore side, and I have not investigated in any depth what happens to the Kelvin wave and the waves of the continuous part of the spectrum. Then, no one reviewed *my* contribution, so I am the one with no safety net here...

I agree writing this amount of text is not reasonable in a review, and I would perfectly understand the authors to consider I'm going too far. Still, with such complexities lurking in the system, I can not agree with their statement that they have "unequivocally identif(ied) CTW mode scattering within observations" (their line 235). I'm afraid there are too many basic things about this problem that are not well understood for unequivocal statements about scattering to be made...

Please also note the supplement to this comment:
https://os.copernicus.org/preprints/os-2020-46/os-2020-46-RC2-supplement.pdf

**Supplement:**

**Choosing a scalar product to make the CSW propagation operator anti-self-adjoint.**

An anonymous referee

September 6, 2020

**Abstract**

My issue with the use of C-EOFs in such a problem is that the correlation matrix that is diagonalized is by construction Hermitian. As is well know, this entails that its eigenmodes are *always* orthogonal for the euclidean scalar product, whereas the standard (Huthnance 1975) paper states very clearly that CSW modes are *not* orthogonal for this scalar product (and there is little hope the introduction of stratification should make things any easier in this respect). In my opinion, this means that considering the successive EOF eigenmodes (orthogonal by construction) as in one-to-one correspondance with the CSW modes (classically not orthogonal) may happen to be justifiable by a careful analysis, but is definitely not a trivial result... In this text I show that a scalar product can be found for which the different CSW modes are orthogonal. This scalar product involves the sea surface elevation perturbation, which does not seem available to (Brunner and Lwiza 2020, subm.).

**1. Basic linear algebra results.**

**1a.** *For matrices*

The argument is very classic, I reproduce it here for the sake of completeness. Let a $T$ superscript denote the transpose of a matrix, an asterisk the element-wise complex conjugate of a matrix, and an $H$ superscript the Hermitian conjugate of a matrix, obtained by transposing its element-wise complex conjugate. Let $\mathbf{M}$ be a Hermitian matrix, *i.e.* such that $\mathbf{M}^H = (\mathbf{M}^T)^* = \mathbf{M}$. Let $\mathbf{X}$ and $\mathbf{Y}$ be two of its eigenvectors, associated respectively to the eigenvalues $\lambda$ and $\mu$.

Then a first series of manipulations shows:

$$\mathbf{X}^H \mathbf{M} \mathbf{X} = \lambda \mathbf{X}^H \mathbf{X} = \lambda ||\mathbf{X}||^2.$$

Then

$$(\mathbf{X}^H \mathbf{M} \mathbf{X})^H = \lambda^* ||\mathbf{X}||^2,$$

but in the meantime

$$(\mathbf{X}^H \mathbf{M} \mathbf{X})^H = \mathbf{X}^H \mathbf{M}^H \mathbf{X} = \mathbf{X}^H \mathbf{M} \mathbf{X} = \lambda ||\mathbf{X}||^2$$

hence:

$$\lambda^* ||\mathbf{X}||^2 = \lambda ||\mathbf{X}||^2.$$

Any eigenvalue of an hermitian matrix is thus real.

In a second stage, one sees that:

$$\mathbf{X}^H (\mathbf{M} \mathbf{Y}) = (\mathbf{M} \mathbf{X})^H \mathbf{Y}$$

entails

$$\mu \mathbf{X}^H \mathbf{Y} = \lambda^* \mathbf{X}^H \mathbf{Y}$$

hence, as $\lambda^* = \lambda$:

$$[\mu - \lambda]\,\mathbf{X}^H\mathbf{Y} = 0.$$

Any two eigenvectors of an hermitian matrix are thus orthogonal, or associated to the same eigenvalue.

This last result carries over to the case of an anti-Hermitian matrix (*i.e.* such that $\mathbf{M}^H = -\mathbf{M}$), though one can show that any eigenvalue $\lambda$ is in this case pure imaginary.

As the matrix that is analyzed in the C-EOF method is by construction Hermitian, the EOF modes are *always* orthogonal, and the method is not adapted to the analysis of a system in which the main modes of motion have no reason to (or have good reasons not to) be orthogonal.

**1b.** *For linear operators*

The considerations above carry quite straightforwardly over from the matrix setting to the linear differential operators setting. In this case, one considers a scalar product, denoted as $\langle \cdot ; \cdot \rangle$ and a linear operator $\mathcal{L}$ operating on a functional space. Then one considers the ajoint linear operator, $\mathcal{L}^+$, such that for all fields $x$ and $y$ in suitable functional spaces:

$$\left\langle \mathcal{L}^+ x; y \right\rangle = \langle x; \mathcal{L} y \rangle$$

Then there is a close correspondance between the properties of an Hermitian matrix and those of a self-adjoint operator (*i.e.* such that $\mathcal{L}^+ = \mathcal{L}$), and between the properties of an anti-Hermitian matrix and those of an anti-self-adjoint operator (*i.e.* such that $\mathcal{L}^+ = -\mathcal{L}$).

**2.  Changing the scalar product**

One thing one should be careful about, though, is that the adjoint of an operator depends on the scalar product used. An operator that is not self-adjoint for a given scalar product can sometimes be made self-adjoint by finding a scalar product that "suits" it better. The orthogonality property of eigenvectors associated to different eigenvalues is then satisfied for the scalar product for which the operator is self-adjoint. Since in the original (Huthnance 1975) paper the phase velocities of CSWs are said to be purely real, it seems there is hope that another choice of scalar product could make the operator either self-adjoint or anti-self-adjoint. Finding a way to insert the discretized version of this scalar product in the C-EOF matrix could allow a more convincing separation of the different CSW modes.

**2a.** *In the CSW problem*

In the notations of the (Huthnance 1975) paper, the equations of motion are summarized as (with $D^2$ a constant):

$$\partial_t \begin{vmatrix} u \\ v \\ \zeta \end{vmatrix} = \begin{bmatrix} 0 & 1 & -\partial_x \\ -1 & 0 & -\partial_y \\ -\frac{1}{D^2}\partial_x(h\cdot) & -\frac{1}{D^2}\partial_y(h\cdot) & 0 \end{bmatrix} \begin{vmatrix} u \\ v \\ \zeta \end{vmatrix}, \tag{1}$$

with $hu \to 0$ for $x \to 0$, $\zeta \to 0$ for $x \to \infty$ as boundary conditions.

Heuristically, I find it convenient to introduce a change of variables to bring the matrix on the right-hand-side as close as possible to anti-self-adjoint form (dividing $\zeta$ by $h$ probably brings in complications close to the shore).

$$\partial_t \begin{vmatrix} u \\ v \\ \zeta\frac{D}{h} \end{vmatrix} = \begin{bmatrix} 0 & 1 & -\frac{1}{D}\partial_x(h\cdot) \\ -1 & 0 & -\frac{1}{D}\partial_y(h\cdot) \\ -\frac{1}{Dh}\partial_x(h\cdot) & -\frac{1}{Dh}\partial_y(h\cdot) & 0 \end{bmatrix} \begin{vmatrix} u \\ v \\ \zeta\frac{D}{h} \end{vmatrix}. \tag{2}$$

With this change of variables, the equations of motion can be written as:

$$\partial_t \left( \mathbf{u}, \frac{D}{h}\zeta \right) = \mathcal{L}\left( \mathbf{u}, \frac{D}{h}\zeta \right),$$

with

$$\mathcal{L} : (\mathbf{B}, g) \longrightarrow \left( -\mathbf{k} \wedge \mathbf{B} - \frac{1}{D}\mathrm{grad}(hg), -\frac{1}{Dh}\mathrm{div}(h\mathbf{B}) \right).$$

Finding a scalar product for which $\mathcal{L}$ is self-adjoint amounts to finding a scalar product for which,

$$\forall (\mathbf{A}, f), (\mathbf{B}, g), \langle (\mathbf{A}, f); \mathcal{L}(\mathbf{B}, g) \rangle = \langle \mathcal{L}(\mathbf{A}, f); (\mathbf{B}, g) \rangle$$

Taking as scalar product

$$\langle (\mathbf{A}, f); (\mathbf{B}, g) \rangle = \int_{\mathcal{D}} \left[ h\mathbf{A}^* \cdot \mathbf{B} + h^2 f^* g \right] dS,$$

where the integration is over the complete study domain, one can perform the following manipulations:

$$
\begin{aligned}
\langle (\mathbf{A}, f); \mathcal{L}(\mathbf{B}, g) \rangle &= \int_{\mathcal{D}} \left[ h\mathbf{A}^* \cdot \left[ -\mathbf{k} \wedge \mathbf{B} - \frac{1}{D}\mathrm{grad}(hg) \right] + h^2 f^* \left[ -\frac{1}{Dh}\mathrm{div}(h\mathbf{B}) \right] \right] dS \\
&= \int_{\mathcal{D}} \left[ h\mathbf{B} \cdot (\mathbf{k} \wedge \mathbf{A}^*) - \frac{1}{D} h\mathbf{A}^* \cdot \mathrm{grad}(hg) - \frac{1}{D} hf^* \mathrm{div}(h\mathbf{B}) \right] dS \\
&= \int_{\mathcal{D}} \left[ h\mathbf{B} \cdot (\mathbf{k} \wedge \mathbf{A}^*) + \frac{1}{D} hg\,\mathrm{div}(h\mathbf{A}^*) + \frac{1}{D} h\mathbf{B} \cdot \mathrm{grad}(hf^*) \right] dS \\
&\quad - \frac{1}{D} \int_{\mathcal{D}} \left[ \mathrm{div}(h^2 f^* \mathbf{B}) + \mathrm{div}(h^2 \mathbf{A}^* g) \right] dS \\
&= \int_{\mathcal{D}} \left[ h\mathbf{B} \cdot \left[ \mathbf{k} \wedge \mathbf{A}^* + \frac{1}{D}\mathrm{grad}(hf^*) \right] + h^2 g \left[ \frac{1}{Dh}\mathrm{div}(h\mathbf{A}^*) \right] \right] dS \\
&\quad - \frac{1}{D} \int_{\mathcal{D}} \left[ \mathrm{div}(h^2 f^* \mathbf{B}) + \mathrm{div}(h^2 \mathbf{A}^* g) \right] dS \\
&= -\langle \mathcal{L}(\mathbf{A}, f); (\mathbf{B}, g) \rangle - \frac{1}{D} \int_{\mathcal{D}} \left[ \mathrm{div}(h^2 f^* \mathbf{B}) + \mathrm{div}(h^2 \mathbf{A}^* g) \right] dS \\
&= -\langle \mathcal{L}(\mathbf{A}, f); (\mathbf{B}, g) \rangle - \frac{1}{D} \int_{\partial\mathcal{D}} \left[ h^2 f^* \mathbf{B} + h^2 \mathbf{A}^* g \right] \cdot d\mathbf{n}
\end{aligned}
$$

It seems reasonable (though it probably should be checked) that the boundary term in this last equation vanishes for fields satisfying the boundary conditions imposed on the flow (one probably should use periodicity in the along-shore direction, and should check that the cross-shore transport vanishes fast enough at the coast, and that the of sea surface height perturbation vanishes fast enough far from the coast. I suspect trouble arises for the Kelvin wave). If this is the case, we see that for this scalar product the operator governing the motion of the CSWs in this framework is anti-self-adjoint, and that its eigenmodes are orthogonal. Namely, if $(u_n, v_n, \zeta_n)$ and $(u_m, v_m, \zeta_m)$ are two CSW modes with different phase velocities

$$\int_{\mathcal{D}} \left[ h(u_n^* u_m + v_n^* v_m) + D^2 \zeta_n^* \zeta_m \right] dS = 0$$

this is consistent with proposition (m,ii) of (Huthnance 1975) that the energy defined in this way separates gracefully in contributions from the different CSW modes.

**2b. *For a matrix diagonalization problem**

The eigenmodes obtained by the C-EOF method are orthogonal with respect to the euclidean scalar product, which is a discretized version of the scalar product

$$\langle f; g \rangle = \int_D f^* g\, dS,$$

where the integration domain is restricted to the domain where observations are available, and a weighting proportional to the density of observations is implicitly applied. Considering the simple case where we want the eigenvectors to be orthogonal with respect to a scalar product that has a different (but still local in space) weighting function $w$,

$$\langle f; g \rangle = \int_D f^* g w(\mathbf{x}) dS,$$

it seems the least one can do is to multiply the empirical covariance matrix to the left and to the right by the diagonal matrix containing the square root of the weighting function at the observation point $\mathbf{W^{1/2}}$, diagonalize

$$\mathbf{P} = \mathbf{W^{1/2}} \mathbf{M} \mathbf{W^{1/2}}$$

instead of $\mathbf{M}$, then multiply the eigenvectors by the inverse of $\mathbf{W^{1/2}}$, and call *these vectors* the empirical eigenmodes.

In this setting:
– the scalar product of two vectors $\mathbf{X}$ and $\mathbf{Y}$ is equal to $\mathbf{X}^H \mathbf{W} \mathbf{Y}$.
– if $\mathbf{X}$ and $\mathbf{Y}$ are two eigenvectors of $\mathbf{P}$ associated to different eigenvalues, they are orthogonal with respect to the Euclidean scalar product, $\mathbf{X}^H \mathbf{Y}$ (which is not the scalar product we want).
– but then $\mathbf{W^{-1/2}} \mathbf{X}$ and $\mathbf{W^{-1/2}} \mathbf{Y}$ are such that:

$$(\mathbf{W^{-1/2}} \mathbf{X})^H \mathbf{W} (\mathbf{W^{-1/2}} \mathbf{Y}) = \mathbf{X}^H \mathbf{W^{-1/2}} \mathbf{W} \mathbf{W^{-1/2}} \mathbf{Y} = \mathbf{X}^H \mathbf{Y} = 0$$

This shows that by scaling the observations in an appropriate way one can obtain eigenvectors of the covariance matrix which are orthogonal with respect to a non-trivial scalar product.

**3. So what?**

- Maybe the empirical eigenmodes retrieved in this way could be in better agreement with the theoretical ones?

- Is there still need for higher-order eigenmodes to explain the variance, or has the "scattering" decreased in importance?

Clearly, the authors lack sea surface height observations:

- Maybe for this type of waves the sea surface elevation signature can be derived from the surface currents?

- Maybe for this type of waves it is a negligible contribution to the scalar product?

I have not given these issues any thoughts. But the impact of the weighting by the water column height is easy to test, and I'd be surprised if it was negligible.

**References**

Brunner, K. and K. Lwiza, 2020, subm.: Evidence of coastal trapped wave scattering using high-frequency radar data in the Mid-Atlantic Bight. *Ocean Science Discussion*.

Huthnance, J. M., 1975: On trapped waves over a continental shelf. *J. Fluid Mech.*, **69**, 689–704.

---

## Referee Comment (RC3) · Anonymous Referee #2 · 10 Sep 2020

Summary: This is an interesting paper that uses a year of HF radar surface currents from the Mid-Atlantic Bight to examine the potential for coastal trapped waves over the continental shelf. The MAB is a broad, flat shelf with significant HFR coverage and is likely an ideal place to do this. I've not seen other studies that directly examine CTWs within HFR results, and thus the work is novel for this reason alone.

However, the analysis hinges on use of EOF modes as direct representations of dynamical modes of CTW variability. The linkages between EOF modes to CTW modes are simply assumed a priori and never significantly tested. The one effort to do so, comparing the across-shelf structure of modes to simple modal theory for free CTWs

is described as poor. A second assumption that breaks in the c-EOF phase or a decrease in local amplitude is unequivocal proof of 'scattering' into higher modes is also not well justified. Unfortunately these two assumptions underlay the bulk of the analysis described. The O(1) influence of the local wind in forcing surface flows appears to be largely ignored.

As I said, I've not seen very much HFR analysis of CTWs in the literature. I'd assumed that this was due to a mismatch of timing; big HFR arrays have only been operational for the past decade (but their data quality is always suspect), and much of the CTW work was done in the 1980s to 2000. I think this type of effort would be great to see more of, but this present work appears to rest on assumptions that need significant support.

Other Major comments

The propagation of features down the wave guide is a significant part of CTW theory. Propagation is never confirmed here. Additionally, data analysis textbooks often show that propagating plane waves can be separated into different modes of variability within an EOF-type decomposition. Is this happening here?

By eye the mode 1 R-EOF amplitude time series has a non-zero mean (Fig 4g), which would suggest a background time-mean flow in the EOF mode itself, is this the case? I'd think that, as the time mean is removed from the EOF calculation, that the time mean of the mode should also be zero.

Given the uniform distribution of mode 1 spatial structure and its slow 'rotation' over time, isn't this just the influence of the time-varying large scale wind field (which is predominantly in the 3-12 day band) on the surface currents?

Winds over the MAB are not completely uniform. The likely heavy influence of the local winds in forcing surface currents at the same frequency bands is not investigated. How are each mode related to the local winds? There are enough buoy observations in the

[Figure]

MAB to test this. I'll suggest that an alternative interpretation of the difference between winter and summer mode 2 results are instead due to changes in the local wind forcing.

Do you think these are free or forced CTWs? This would be important for both the propagation and 'scattering' effects. Additionally, given that the MAB is a broad, shallow shelf, the role of friction on CTW should be significant as the size of the waves is long relative to the along-shelf distance. Can this be addressed in your comparisons?

Minor comments

Line 55: Optimal interpolation applies some modal analysis to determine vectors from the radials. What role might this play in pre-conditioning the EOF calculation?

Line 65: The location of the buoy is not defined.

Equ 1-5, if you are going to present the governing equations in this detail, you might also wish to discuss each equation and/or relate the results at the end to this presentation, otherwise, is it needed information?

Line135: This sentence contradicts itself.

Line 140: Again, jumping to scattering as the reason for this difference seems presumptuous. What might the role of plume dynamics downstream of the Del and Ches. Bays play on the across-shelf structure of along shelf currents? While Zhang and Lentz worked specifically on the HC area, Zhang's earlier work also showed a natural break in circulation at the canyon due to wind driven circulation.

Line 155: The sharp discontinuity that exists along the eastern edge of Cape Cod (for both modes 1 and 3) is somewhat troubling. There is no bathymetric feature, or coastline orientation that might align to explain the scattering. Knowing the MARACOOS array, I know that one of their radars is located exactly at the location of the discontinuity, raising the possibility that this aspect of the results is due to instrumentation issues.

[Figure]

Figure 4 and 5: You are presenting the results as along and across-shelf, but interpreting the winds as zonal and meridional. This does not allow for an easy determination of forced vs. free modes. The changes in coastline orientation does not necessarily preclude this, as the CTW's should propagate southward, and be most affected by upstream winds.

Line 168: I'm not sure I understand the argument here. You are suggesting that increased variance in higher modes suggests that 'scattering' of low modes into high modes is occurring. yet, the Mode 3 energy at the HSV is low, not high. . .would this suggest that scattering is not occurring at this location?

Line 175: By extension, this statement suggests you believe that EOF modes 20-30 are also representative of CTWs, is this the case? If not at which modal number to you think your assumption of EOF mode = CTW mode breaks down?

Line 180: why would freshwater inflow induce scattering into higher dynamical modes?

Line 185: You are suggesting that the summer mode 1 does not feel the canyon due to stratification. This would also suggest that, if it is an CTW, it is a baroclinic mode 1 wave, which would have a dramatically slower, and measurable phase speed. Is this observed in the data?

Line 190: The CTW velocities are band pass filtered between 3 and 12 days. What about the energy in the 0 to 3 day and 12 to monthly energy bands?

Line 214: It is not clear what part of the vector field in figures 7c and 8c are representative of wave reflection?

Line 215: please refer to fig9 here.

Line 217: Is this assumption also true in HSV? I ask as you seem to make the opposite argument there. . . Why are they different?

Line 234: I disagree with your use of the term 'unequivocally'. In the previous section,

you ascribe spatial variations that are occurring in the same area as the Chesapeake outflow plume as due to CTW just because they occur in the 3-12 day band. This is not proof of CTW scattering.

Line 237: I disagree that you have the 'necessary framework to demonstrate that scattering was occurring'.

Line 240: '...large, sudden jumps in phase on the C-EOF phase maps are indications of scattering...' What other processes might cause the same effects?

Line 244: See my above comment...the sharp change in phase/amplitude is not in an area of strong change in bathymetry. Why else would CTWs be changing here? Additionally, the discontinuity causes divergence and convergence, depending on the sign of the amplitude timeseries, not just divergence.

Line 260: Its not obvious how you are defining reflections from an observational point of view. What evidence suggests this in the data?

---

## Author Comment (AC1) · 8 Oct 2020

Response to the first comment by Reviewer 2:

Point 1:
The reviewer states that:
**"However, the analysis hinges on use of EOF modes as direct representations of dynamical modes of CTW variability. The linkages between EOF modes to CTW modes are simply assumed a priori and never significantly tested. The one effort to do so, comparing the across-shelf structure of modes to simple modal**

[Figure]

**theory for free CTWs."**
This has already been addressed in response to Reviewer 1 comments.

Point 2:
**A second assumption that breaks in the c-EOF phase or a decrease in local amplitude is unequivocal proof of 'scattering' into higher modes is also not well justified.**
Please see Wang (1980) as cited for a more greater detailed explanation of how sudden changes in phase and amplitude are indications of scattering. We agree that our use of the word "unequivocal" may be a bit strong and will re-word.

Point 3:
**big HFR arrays have only been operational for the past decade (but their data quality is always suspect)**
We beg to differ. HFR data quality has improved considerably. See Brunner  Lwiza (2020), which uses the data to compute tidal velocities and residual velocities. Also, the US Coast Guard now uses HFR data for search and rescue operations.

Point 4:
**The propagation of features down the wave guide is a significant part of CTW theory. Propagation is never confirmed here. Additionally, data analysis text-books often show that propagating plane waves can be separated into different modes of variability within an EOF-type decomposition. Is this happening here?**
This is introducing a different avenue which is not our focus.

Point 5:
**By eye the mode 1 R-EOF amplitude time series has a non-zero mean (Fig 4g), which would suggest a background time-mean flow in the EOF mode itself, is this the case? I'd think that, as the time mean is removed from the EOF**

**calculation, that the time mean of the mode should also be zero.**
Unless we made a mistake the time mean should be zero as the data is demeaned prior to EOF analysis. Regardless, we will check this.

Point 6:
**'Given the uniform distribution of mode 1 spatial structure and its slow 'rotation' over time, isn't this just the influence of the time-varying large scale wind field (which is predominantly in the 3-12 day band) on the surface currents?'**
We will plot wind versus mode 1 to investigate this point.

Point 7:
**'the difference between winter and summer mode 2 results are instead due to changes in the local wind forcing'**
The reviewer may be right. We need to test this, although it may be difficult to separate the influence of changing wind forcing and stratification.

Point 8:
**Do you think these are free or forced CTWs? This would be important for both the propagation and 'scattering' effects.**
These should be the free waves that propagate once the forcing has relaxed.
**Additionally, given that the MAB is a broad, shallow shelf, the role of friction on CTW should be significant as the size of the waves is long relative to the along-shelf distance. Can this be addressed in your comparisons?**
Theoretical frictional decay scales calculated by the Brink model are generally less than one day for mode 1 and on the order of several days for higher order modes. Relative to the period of the waves, in addition to their fast phase speed, friction is not significant in this region.

Point 9:

**Line 55: Optimal interpolation applies some modal analysis to determine vectors from the radials. What role might this play in pre-conditioning the EOF calculation**
We were not aware of this, but it can easily be proved by comparing it to linear interpolation.

Point 10:
**Line 65: The location of the buoy is not defined.**
This is a good point. It will be rectified.

Point 11:
**Equ 1-5, if you are going to present the governing equations in this detail, you might also wish to discuss each equation and/or relate the results at the end to this presentation, otherwise, is it needed information?**
The governing equations are important to present in this manner as they are the underlying equations for the Brink model, in addition to the general physics. Furthermore, with the argument for orthogonality using Wang Mooers (1976) formula, we need the equations even more.

Point 12:
**Line135: This sentence contradicts itself.**
It does not contradict itself, but we will elaborate in the revised version.

Point 13:
**Line 140: Again, jumping to scattering as the reason for this difference seems presumptuous. What might the role of plume dynamics downstream of the Del and Ches. Bays play on the across-shelf structure of along shelf currents? While Zhang and Lentz worked specifically on the HC area, Zhang's earlier work also showed a natural break in circulation at the canyon due to wind driven**

**circulation.**
This is discussed in the manuscript and Brunner Lwiza (2019).

Point 14:
**Line 155: The sharp discontinuity that exists along the eastern edge of Cape
Cod (for both modes 1 and 3) is somewhat troubling. There is no bathymetric
feature, or coastline orientation that might align to explain the scattering.
Knowing the MARACOOS array, I know that one of their radars is located exactly
at the location of the discontinuity, raising the possibility that this aspect of the
results is due to instrumentation issues.**
See Brunner Lwiza (2020). We agree that it is unusual, but other observations and
numerical model results also show that discontinuity.

Point 15:
**Figure 4 and 5: You are presenting the results as along and across-shelf, but
interpreting the winds as zonal and meridional. This does not allow for an easy
determination of forced vs. free modes. The changes in coastline orientation
does not necessarily preclude this, as the CTW's should propagate southward,
and be most affected by upstream winds.**
Point well taken. We took the winds to represent the whole region, but will rotate the
winds to along and across-shelf and add data from other buoys in the region.

Point 16:
**Line 168: I'm not sure I understand the argument here. You are suggesting that
increased variance in higher modes suggests that 'scattering' of low modes into
high modes is occurring. yet, the Mode 3 energy at the HSV is low, not high. .
.would this suggest that scattering is not occurring at this location?**
It is not clear to us what is happening in HSV. Probably not scattering, per se,
but it could be absorption. When you watch videos of the CTW activity, the signal

propagating from the north dies out when it reaches the HSV, and then reappears to the south on the New Jersey side.

Point17:

**Line 175: By extension, this statement suggests you believe that EOF modes 20-30 are also representative of CTWs, is this the case? If not at which modal number to you think your assumption of EOF mode = CTW mode breaks down?**
We do not know for sure, but probably modes 20-30 represent  0.01

Point18:
**Line 180: why would freshwater inflow induce scattering into higher dynamical modes?**
That is what we observe, we do not know exactly why. It is one of the many questions that need further investigation.

Point 19:
**Line 185: You are suggesting that the summer mode 1 does not feel the canyon due to stratification. This would also suggest that, if it is an CTW, it is a baroclinic mode 1 wave, which would have a dramatically slower, and measurable phase speed. Is this observed in the data?**
We do not observe a dramatically slower wave with measurable phase speed in summer. As stated above, there appears to be more absorption than scattering at this location that may also be explanation for this unusual behavior.

Point 20:
**Line 190: The CTW velocities are band pass filtered between 3 and 12 days. What about the energy in the 0 to 3 day and 12 to monthly energy bands?**
The 0-3 day bandwidth is dominated by tides, and based on wavelet analysis 12-day to monthly bandwidth contains very little CTW energy.

[Figure]

Point 21:

**Line 214: It is not clear what part of the vector field in figures 7c and 8c are representative of wave reflection?**

When the phase has sudden change (almost reversed).

Point 22:

**Line 215: please refer to fig9 here.**

We agree – we will.

Point 23:

**Line 217: Is this assumption also true in HSV? I ask as you seem to make the opposite argument there. . . Why are they different?**

The assumption holds for most parts except HSV, and we do not know why. Observations do not always agree with theory. Further work beyond the scope of this paper needs to be done to find out why.

Point 24:

**Line 234: I disagree with your use of the term 'unequivocally'. In the previous section, you ascribe spatial variations that are occurring in the same area as the Chesapeake outflow plume as due to CTW just because they occur in the 3-12 day band. This is not proof of CTW scattering.**

We agree that the word "unequivocal" was too strong. We will rewrite the statement.

Point 25:

**Line 237: I disagree that you have the 'necessary framework to demonstrate that scattering was occurring'.**

We disagree with the reviewer here, because we think we do.

Point 26:

**Line 240: '. . .large, sudden jumps in phase on the C-EOF phase maps are indications of scattering. . .' What other processes might cause the same effects?**

We do not know of any other process that can cause a sudden change in phase like we are observing except scattering.

Point 27:
**Line 244: See my above comment. . .the sharp change in phase/amplitude is not in an area of strong change in bathymetry. Why else would CTWs be changing here? Additionally, the discontinuity causes divergence and convergence, depending on the sign of the amplitude timeseries, not just divergence.**

In the area you are referring to we see the phase jumps even in the tidal signal, and even model results indicate similar features. However, we do not know why. Since it seems an area that brings more confusion to an already complex analysis we will exclude it from the results, because it does not add value.

Point 28:
**Line 260: Its not obvious how you are defining reflections from an observational point of view. What evidence suggests this in the data?**

Reflection by definition is scattering, but we will change the wording here – estuary to freshwater flow.

References:
Brunner, K. and Lwiza, K. M. M., 2020. Tidal velocities on the Mid-Atlantic Bight continental shelf using high-frequency radar. Journal of Oceanography, 76, pp.289-306.

Brunner, K. and Lwiza, K. M. M., 2019. The impact of storm-induced coastal trapped waves on the transport of marine debris using high-frequency radar data. 2019 IEEE/OES Twelfth Currents, Waves and Turbulence Measurement (CWTM),

pp.1-5.

Wang, D.P. and Mooers, C.N., 1976. Coastal-trapped waves in a continuously stratified ocean. Journal of Physical Oceanography, 6(6), pp.853-863.

---

## Author Comment (AC2) · 9 Oct 2020

Thank you Dr. Lwiza for replying to the first comment by Reviewer 1 on our behalf. I will now be responding to the second comment provided by Reviewer 1.

Response to the second comment by Reviewer 1:

Point 1:
The reviewer states that:
**"The MAB has been the subject of very many studies, whether observational**

**of numerical. I'm surprised the authors could not propose any reference to support their view."**
Compared to other subfields of ocean physics, the field of coastal trapped waves has a very thin literature. However, in the manuscript we mention a few studies in the region from lines 37 to 38. There are two other references we did not include in that list, i.e., Zhang and Lentz (2017) and Zhang and Lentz (2018), but are included elsewhere in the manuscript. We agree with the reviewer's point that they should be included in the first list. We do acknowledge that there may be other processes in that bandwidth, but the wavelet analysis shows that their energy is small compared to the coastal trapped waves (CTW). We do agree that waves propagate, but it should be remembered that as they do they are either scattered or absorbed (in lossy media). However, it might not be fool proof, but what our results show is more than anecdotal evidence that waves are being scattered to higher modes. This has never been shown before, and it provides an important contribution to the field.

Point 2:
The reviewer admits oversight, but we also see how one could easily make that mistake based on the way we presented both C-EOF and R-EOF in such a compact form. We will try to separate the statements to emphasize the distinction and elaborate more on the methods. We also agree that it would be desirable to be able to combine the U and V fields, but this is not a paper that will address every challenge that exists in coastal trapped waves. Our major objective is to analyze the data than developing a new method. Hence we used available tools. Modification or development of a new method is left for another study.

Point 3:
This is the main contention the Reviewer has against the manuscript.
It is true that CTW modes are not orthogonal as shown by Huthnance (1975). However, we want to draw the reviewer's attention to the literature showing that the orthogonal

condition for CTW does exist. It includes a bottom slope term, dH/dx (see Eqn. 29a in Wang  Mooers, 1976; the first term can be neglected without loss of generality), whereas the EOF does not. If one ignores the Georges Bank region and the Hudson Canyon, the MAB bottom slope is approximately constant as the data are mostly over the broad shelf. We ran Brink's model to produce the theoretical modes, which were applied in the Wang  Mooers Eqn 29a written as:

$$\int_{-H_0}^0 P_n^0 P_m^0 dz + \int_0^\infty \left(\frac{dH}{dx}\right) P_n^1 P_m^1 dx = 0 \text{ for } n \neq m \text{ (1)}$$

where $H_0$ is depth at the coast, $P_n^0$ is nth mode at the coast, $P_n^1$ is nth mode computed across the shelf. We then discretize (1) and drop the first term as follows:

$$\sum_1^{N_x} P_n^1 P_m^1 = 0 \text{ for } n \neq m \text{ (2)}$$

As expected the shelf being broad keeps the dH/dx term constant and near zero until it reaches the shelf break (see attached Figure 6). Therefore, for most part of the shelf to prove the existence of orthogonality between modes we need to show that:

$$\sum_1^{N_x} P_1^1 P_2^1 \ll \sum_1^{N_x} P_1^1 P_1^1 \text{ and } \sum_1^{N_x} P_1^1 P_2^1 \ll \sum_1^{N_x} P_2^1 P_2^1 \text{ (3)}$$

We used the data for the transects shown in attached Figure 1, except MA, to compute $\sum_1^{N_x} P_1^1 P_2^1 / \sum_1^{N_x} P_1^1 P_1^1$ and $\sum_1^{N_x} P_1^1 P_2^1 / \sum_1^{N_x} P_2^1 P_2^1$, with the expectation that both will be much less than 1. Attached Figures 2 – 5 show the first three theoretical modes from Brink's model in the top panel and the bottom panel shows the plots of the modes multiplied by the dH/dx following Equation 29a of Wang  Mooers (1976). The ratios are less than one in magnitude, but only the Delaware transect has absolute values which are much less than one. This could be because the real topography that is used is introducing errors of deviation from orthogonality. This is an area that needs to be investigated further; for instance, there seems to be a need of smoothing the bathymetry before calculating the modes. Another source of discrepancy may stem from the fact that winds over the study area are not entirely uniform as is the assumption in Brink's model. In any case, we have one transect that satisfies orthogonality which we can use to examine the theoretical modes in relation to the EOF modes.

References:

Huthnance, J. M., 1975: On trapped waves over a continental shelf. J. Fluid Mech., 69, 689-704.

Wang, D.P. and Mooers, C.N., 1976. Coastal-trapped waves in a continuously stratified ocean. Journal of Physical Oceanography, 6(6), pp.853-863.

[Figure]

Figure 1. Transects for Brink's CTW mode analysis (blue lines). Solid black lines indicate controlled HFR coverage and light grey lines are bathymetric contours.

**Fig. 1.**

[Figure]

Figure 2. Hudson Canyon transect Brink's modes (top) and the modes multiplied by dH/dx term (bottom). sum(p1*p2)/sum(p1*p1)=-0.2063, sum(p1*p2)/sum(p2*p2)=-0.6174.

**Fig. 2.**

[Figure]

Figure 3. Same as Figure 2 for New Jersey transect. sum(p1*p2)/sum(p1*p1)=-0.2683, sum(p1*p2)/sum(p2*p2)=-0.4701.

**Fig. 3.**

[Figure]

Figure 4. Same as Figure 2 for Delaware transect. sum(p1*p2)/sum(p1*p1)=-0.0007, sum(p1*p2)/sum(p2*p2)=-0.0031.

**Fig. 4.**

[Figure]

Figure 5. Same as Figure 2 for Delaware transect. sum(p1*p2)/sum(p1*p1)=-0.0961, sum(p1*p2)/sum(p2*p2)=-0.5008.

**Fig. 5.**

[Figure]

Figure 6. dH/dx term for the New Jersey transect. All transects have a similar (not exact) shape showing a relatively flat shelf.

**Fig. 6.**